# Risk of Gestational Diabetes and Pregnancy-Induced Hypertension with a History of Polycystic Ovary Syndrome: A Nationwide Population-Based Cohort Study

**DOI:** 10.3390/jcm12051738

**Published:** 2023-02-21

**Authors:** Seung-Woo Yang, Sang-Hee Yoon, Myounghwan Kim, Yong-Soo Seo, Jin-Sung Yuk

**Affiliations:** Department of Obstetrics and Gynecology, Sanggye Paik Hospital, School of Medicine, Inje University, Seoul 01757, Republic of Korea

**Keywords:** polycystic ovary syndrome, gestational diabetes, pregnancy-induced hypertension, East Asian population, national cohort study

## Abstract

Objective: To evaluate the risks of developing gestational diabetes (GDM) and pregnancy-induced hypertension (PIH) in women with polycystic ovary syndrome (PCOS) using data from Korea’s National Health Insurance Service. Method: The PCOS group comprised women aged 20 to 49 years diagnosed with PCOS between 1 January 2012, and 31 December 2020. The control group comprised women aged 20 to 49 years who visited medical institutions for health checkups during the same period. Women with any cancer within 180 days of the inclusion day were excluded from both the PCOS and control groups, as were women without a delivery record within 180 days after the inclusion day, as well as women who visited a medical institution more than once before the inclusion day due to hypertension, diabetes mellitus (DM), hyperlipidemia, DM in pregnancy, or PIH. GDM and PIH were defined as cases with at least three visits to a medical institution with a GDM diagnostic code and a PIH diagnostic code, respectively. Results: Overall, 27,687 and 45,594 women with and without a history of PCOS experienced childbirth during the study period. GDM and PIH cases were significantly higher in the PCOS group than in the control group. When adjusted for age, SES, region, CCI, parity, multiple pregnancies, adnexal surgery, uterine leiomyoma, endometriosis, PIH, and GDM, an increased risk of GDM (OR = 1.719, 95% CI = 1.616–1.828) was observed among women with a history of PCOS. There was no increase in the risk of PIH among women with a history of PCOS (OR = 1.243, 95% CI = 0.940–1.644). Conclusion: A history of PCOS itself might increase the risk of GDM, but its relationship with PIH remains unclear. These findings would be helpful in the prenatal counseling and management of patients with PCOS-related pregnancy outcomes.

## 1. Introduction

Polycystic ovary syndrome (PCOS) affects 8–13% of women of reproductive age [1] and is associated with dysfunctional gonadotropin secretion [2]. In terms of clinical implications, infertility caused by chronic ovulatory dysfunction, abnormal gonadotropin secretion, and metabolic disturbances, such as central obesity, dyslipidemia, insulin resistance, and hyperinsulinemia, can coincide with PCOS [3].

Women with PCOS have increased risks of pregnancy and delivery complications. A woman with a PCOS-affected pregnancy is more likely to have increased oxidative stress and experience infertility requiring assisted conception [4,5,6]. Normal pregnancy induces a state of insulin resistance that may manifest as impaired glucose tolerance or gestational diabetes (GDM) [7]. Because women with PCOS have a reported incidence of 25–70% of insulin resistance, they would appear to be at increased risk of developing GDM complications [8].

Several previous meta-analyses have suggested that PCOS influences the development of GDM and pregnancy-induced hypertension (PIH) [5,9,10,11]. The early meta-analysis of Boosma et al. found that PCOS-affected pregnancy was associated with significantly higher risks of developing GDM, gestational hypertension, and preeclampsia (PE) [9]. The largest meta-analysis was reported in 2019 and included over 224,000 pregnant women [11], and suggested that PCOS, independent of obesity, increases the risks of developing GDM and PIH. However, the fundamental limitations of meta-analysis meant significant heterogeneity in the included samples, primarily due to differing study designs and ethnic backgrounds. Therefore, further evaluations of PCOS as an independent risk factor for GDM and PIH with comprehensive adjustments for confounding factors are still needed.

A significant finding of recent studies is that PCOS is an independent risk factor for worse birth outcomes. A large population-based cohort study that included 9.1 million births in women with PCOS found that across all pregnancies, women with PCOS were 2-fold and 1.38-fold more likely to develop GDM and PIH, respectively, after controlling for obesity, IVF use, and other confounders [12]. However, that study was characterized by considerable heterogeneity in ethnicity and the sizes of the included samples (14,882 PCOS patients vs. 9,081,906 controls) and did not control for PIH confounding factors such as parity, multifetal pregnancy, and GDM itself. Additionally, there are considerable ethnicity variations in the manifestation of PCOS, with a low body mass index (BMI) and mild hirsutism in East Asian women with PCOS compared with Western and South Asian women with PCOS [13,14]. Therefore, the purpose of the present study was to determine the associations of PCOS with GDM and PIH in Korean women of reproductive age using nationwide population-based data.

## 2. Materials and Methods

### 2.1. Database

Single-payer healthcare is provided to most of the Korean population. Korea’s National Health Insurance is mandatory for all residents [15]. The National Health Insurance Service of Korea provides information about medical records, including gender, age, disease name, income level, kind of medical insurance, name of prescription medicine, surgery received, and hospitalizations. The Health Insurance Review and Assessment Service (HIRA) is an independent organization that evaluates the appropriateness of medical bills to avoid disputes between the National Health Insurance Corporation and medical institutions about insurance premium payments. As a result, the HIRA holds most of the National Health Insurance Corporation’s medical record information. This population-based retrospective cohort study analyzed HIRA health insurance data collected from 1 January 2011 to 31 December 2020.

### 2.2. Selection of Participants

Subject selection and outcome confirmation were based on the ICD-10 and Korea Health Insurance Medical Care Expenses (2016 and 2019 versions). The PCOS group was drawn from women aged 20 to 49 years diagnosed with PCOS between 1 January 2012 and 31 December 2020. Throughout the research period, the clinical guidelines of the Korean Society of Gynecologic Endocrinology suggested only applying the Rotterdam 2003 criteria for diagnosing PCOS [16]. Only women with PCOS who visited medical institutions at least three times with an ICD-10 code of E28.2 were included in the study. The inclusion day was the first PCOS-related visit to a medical institution. The control group comprised women aged 20 to 49 years who visited medical institutions for health checkups between 1 January 2012 and 31 December 2020, excluding women diagnosed with PCOS. If a medical institution was visited at least twice for a health checkup, the date of the initial visit was taken as the inclusion day. Women who visited the hospital for PCOS or a health checkup in 2011 were not eligible to wash out. Women with any cancer (ICD-10 diagnostic code of “any Cxx”) diagnosed within 180 days of the inclusion day were excluded from the PCOS and control groups. Additionally, women without a delivery record within 180 days after the inclusion day were excluded from both groups. Women who visited a medical institution more than once before the inclusion day due to hypertension (HTN) (ICD-10 code = I10~I15), diabetes mellitus (DM) (E10~E14), hyperlipidemia (E78), DM in pregnancy (O24), or PIH (O14~O15) were excluded from both study groups.

### 2.3. Outcome

GDM and PIH were defined as cases with at least three visits to a medical institution with the ICD-10 diagnostic codes for GDM (O24.4) and PIH (O14~O15), respectively.

### 2.4. Variables

When medical aid was the only sort of insurance available to a subject, they were classified as having a low socioeconomic status (SES). If the inclusion region was not urban, it was classified as rural. CCI was determined from the date of inclusion to 1 year prior using diagnostic codes [17]. Parity (primiparity or multiparity) and multiple pregnancies were determined from delivery records. HTN (I10~I15), DM (E10~E14), hyperlipidemia (E78), and obesity (BMI > 25 kg/m^2^, E66) were defined as two or more visits to a medical institution with the associated diagnosis codes. Adnexal surgery was defined as being present when excision of an adnexal tumor, adnexectomy, ovarian wedge resection, or incision and drainage of an ovarian cyst was performed at least once before the inclusion day. Those who visited a medical institution at least twice before the inclusion day with a diagnosis code related to uterine fibroids (D25) or endometriosis (N80) were considered to have the corresponding disease.

### 2.5. Statistical Analyses

SAS Enterprise Guide (version 7.15, SAS Institute, Cary, NC, USA) was used for all statistical analyses, with R software (version 3.5.1, The R Foundation for Statistical Computing, Vienna, Austria) serving as an accessory. A two-sided test was applied in each statistical analysis, and a *p*-value of 0.05 or less was considered statistically significant. Pearson’s chi-square or Fisher’s exact test was used for analyzing categorical variables, and *t*-test or Mann–Whitney *U*-test was used to analyze continuous variables. Logistic regression analysis was used to evaluate the risks of GDM and PIH in PCOS in the presence of certain confounding factors. The inclusion day was chosen as the starting point, and the end date was chosen as the first delivery date after inclusion. The listwise deletion approach was used when there were fewer than 10% missing values, while the regression imputation method was used when there were more than 10% missing values. The validity of our study’s findings was evaluated using logistic regression analysis to determine the risks of GDM and PIH for PCOS in women with a moderate-to-high SES.

### 2.6. Ethics

The IRB of Sanggye Paik Hospital approved this study (approval no.: SGPAIK 2021-12-005). Variables that could be used to identify individuals were de-identified in the study. The analyses were conducted entirely on an offline server provided by HIRA; only calculated numerical values (tables, figures, and numbers) were exportable from the server. This protocol ensured no risk in the study participants being identified. Additionally, the need to obtain informed consent was not required under South Korea’s Bioethics and Safety Act. According to HIRA’s privacy policy, only research results were exportable from the server, meaning raw data cannot be made available to readers. Although this study used HIRA data, HIRA had no interest in the results.

## 3. Results

Patient data on 724,307 women (aged 20–49 years) who underwent a medical checkup or were diagnosed with PCOS during 2012–2020 were extracted from the HIRA database. Extracting data according to delivery status within 180 days after the inclusion day and excluding cancer, HTN, and DM resulted in 73,281 women being enrolled: 45,594 without PCOS and 27,687 with PCOS (Figure 1). Table 1 lists the detailed characteristics of these patients. Their median age was 30 years (interquartile range = 27–33 years). The rate of primiparity was higher in the control group, while the multiple pregnancy rate was higher in the PCOS group. The rate of obesity differed significantly between the two groups.

Table 2 indicates that GDM and PIH incidence rates were higher in the PCOS group than in the control group (GDM: 5.1% vs. 8.4%, *p* < 0.001; PIH: 0.3% vs. 0.4%, *p* = 0.016). PCOS was a risk factor for both GDM (relative risk (RR) = 1.709, 95% CI = 1.610–1.814, *p* < 0.001) and PIH (RR = 1.385, 95% CI = 1.062–1.808, *p* = 0.016). Logistic regression analysis was performed to identify the risk factors of PCOS for GDM and PIH (Table 3). In adjusted logistic regression, PCOS was a risk factor for GDM (RR = 1.719, 95% CI = 1.616–1.828, *p* < 0.001) but not for PIH (RR = 1.243, 95% CI = 0.940–1.644, *p* = 0.127) (Figure 2). However, primiparity (RR = 2.293, 95% CI = 1.292–4.082, *p* = 0.005), multiple pregnancies (RR = 3.668, 95% CI = 2.605–5.165, *p* < 0.001), and endometriosis (RR = 2.399, 95% CI = 1.310–4.395, *p* = 0.005) were significantly associated with PIH.

## 4. Discussion

PCOS is one of the most common endocrine disorders in women of reproductive age [18]. Pregnancy is a diabetogenic state in which progressive insulin resistance that develops during pregnancy due to placental production of diabetogenic hormones decreases glucose entry into maternal cells and preserves fuel for the developing fetus [19]. Therefore, the basic pathophysiologies of insulin resistance and obesity can result in PCOS affecting the progression of the diabetogenic status of pregnancy to adverse metabolic complications, such as androgen excess, dyslipidemia, or low-grade chronic inflammation [20]. Additionally, induced metabolic abnormalities in women with PCOS increase oxidative stress [6]. Therefore, these clinical and biochemical characteristics associated with trophoblast invasion and placentation directly affect pregnancy complications. [21,22] As mentioned above, several previous studies have suggested that PCOS could be an independent risk factor for GDM and PIH. However, the considerable heterogeneity between the studies, including in the ethnic backgrounds of the subjects, means that the relationships remain inconclusive. In the present study, after adjusting confounding factors, PCOS was isolated as a risk factor for GDM, whereas its relationship with PIH remained unclear in a homogeneous East Asian population.

We found that PCOS was associated with an increased risk for GDM of slightly lower than twofold (adjusted RR = 1.719, 95% CI = 1.616–1.828). Compared with previous studies, the present study had a large cohort, and the risk was slightly lower. This difference is probably due to differences in the ethnicity heterogeneity of the included populations and in the confounding factors. Regarding ethnicity, the GDM incidence in this study was affected by the entire included population being Korean. The prevalence of GDM has been estimated at 14% for all pregnancies in the US and has been increasing in multiethnicity populations [23,24]. In contrast, only 2% to 5% of all pregnant Korean women reportedly develop GDM [25]. Regarding confounding factors, the present study was designed to evaluate the risk of PCOS by itself, and so, the analyzed data set excluded a previous history of GDM, PIH, HTN, and DM, which affect multifetal pregnancies, and this would affect the identified risks.

The risk associated with PIH appeared similar to that in a previous study (crude RR = 1.385, 95% CI = 1.062–1.808, *p* = 0.016). However, after adjusting confounding factors, the association was not statistically significant (adjusted RR = 1.243, 95% CI = 0.940–1.644). PIH is a multiorgan disease, so its risk factors are multifactorial [26]. In particular, multifetal pregnancy, nulliparity, and GDM are also suggested as PIH risk factors that were not adjusted in the previous cohort study. Additionally, ethnicity heterogeneity affects PIH development. Table 3 indicates that primiparity and multifetal pregnancy affected the PCOS independent risk presentation. Endometriosis was found to be a risk factor for PIH in our study. Similarly, a recent, large-cohort study suggested that endometriosis affects adverse pregnancy outcomes, including PIH (adjusted RR = 1.17, 95% CI = 1.03–1.33) [27]. Previous research has suggested that women with endometriosis have higher levels of local and systemic inflammation that may influence the risk for specific pregnancy outcomes such as preterm birth, PIH, and pregnancy-induced PE and eclampsia [28,29,30].

The limitation of this study is that it performed a retrospective analysis utilizing an administrative database, which relies on the accuracy and consistency of the individuals coding the data. Obesity is a major clinical presentation of PCOS and a significant risk factor for GDM and PIH. In this study, the incidence of obesity was very low in the control group (0.1%) and the PCOS group (0.2%), which might be due to missing data or inaccurate data coding. Further evaluations with other BMI data, therefore, need to be performed. However, Ryu et al. found only a tiny difference in the obesity rates between normal and PCOS groups in a Korean population (14.4% vs. 15.7%), which was much smaller than that found in the cohort study performed by Mills et al. (3.5% vs. 22.3%) [12,31]. Therefore, although further confirmation is necessary, it appears that obesity did not exert marked effects in the present cohort.

Despite the limitations, the strength of the present study was that it assessed pregnancy risk factors related to PCOS in a large cohort with a single Asian ethnicity. Additionally, multiple confounding factors such as pregestational HTN and DM were excluded, and multifetal pregnancy, parity, and GDM were adjusted to identify the actual risk of PCOS in pregnancy. The incidence of PCOS varies according to the diagnostic criteria employed. PCOS is commonly diagnosed using three diagnostic criteria: NIH criteria, Rotterdam criteria, and Androgen Excess Society criteria [32]. During the study period, the clinical guidelines of the Korean Society of Gynecologic Endocrinology suggested only using the Rotterdam 2003 criteria for diagnosing PCOS, so the methodological heterogeneity was also adjusted. In conclusion, a history of PCOS itself might increase the risk of GDM, but its relationship with PIH remains unclear. Although further studies are needed, the present findings will be helpful in prenatal counseling and managing patients with PCOS-related pregnancy outcomes.

## Figures and Tables

**Figure 1 jcm-12-01738-f001:**
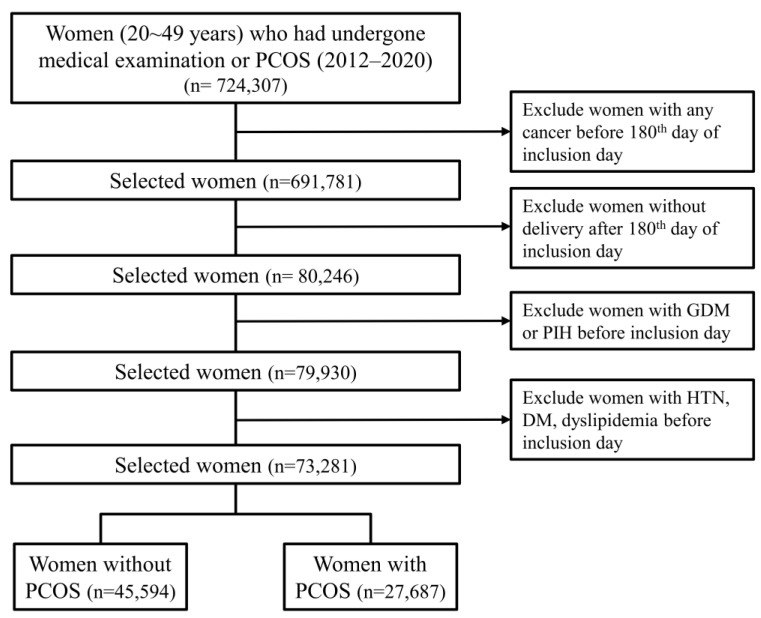
Flowchart of enrollment. DM—diabetes mellitus; GDM—gestational diabetes mellitus; HTN—hypertension; ÍPCOS—polycystic ovary syndrome.

**Figure 2 jcm-12-01738-f002:**
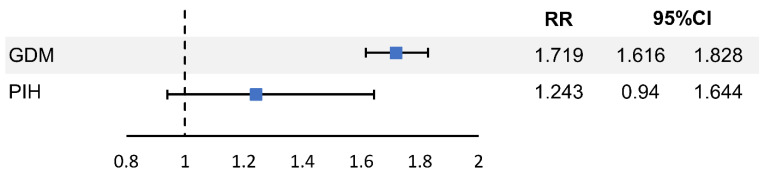
Adjusted relative risks of GDM and PIH in pregnant women with PCOS. GDM—gestational diabetes mellitus; PCOS—polycystic ovary syndrome; PIH—pregnancy-induced hypertension; RR—relative risk.

**Table 1 jcm-12-01738-t001:** From the National Health Insurance database, 2012–2020, baseline characteristics of women with/without PCOS.

	Non-PCOS	PCOS	Total	*p*-Value
Number of women	45,594	27,687	73,281	
Median age (years)	30 (28–33)	29 (27–32)	30 (27–33)	<0.001
Median follow-up period (days)	616 (343–1107)	672 (416–1184)	638 (373–1136)	<0.001
Age at inclusion (years)				<0.001
20~24	3855 (8.5)	3205 (11.6)	7060 (9.6)	
25~29	15,224 (33.4)	10,962 (39.6)	26,186 (35.7)	
30~34	19,463 (42.7)	11,223 (40.5)	30,686 (41.9)	
35~39	6274 (13.8)	2169 (7.8)	8443 (11.5)	
40~44	764 (1.7)	126 (0.5)	890 (1.2)	
45~49	14 (0)	2 (0)	16 (0)	
SES				<0.001
Mid–high SES	45,336 (99.4)	27,591 (99.7)	72,927 (99.5)	
Low SES	258 (0.6)	96 (0.3)	354 (0.5)	
Region				<0.001
Urban area	28,915 (63.4)	14,548 (52.5)	43,463 (59.3)	
Rural area	16,679 (36.6)	13,139 (47.5)	29,818 (40.7)	
CCI				<0.001
0	37,719 (82.7)	23,590 (85.2)	61,309 (83.7)	
1	5569 (12.2)	3138 (11.3)	8707 (11.9)	
≥2	2306 (5.1)	959 (3.5)	3265 (4.5)	
Parity				<0.001
Primi	38,389 (84.2)	25,693 (92.8)	64,082 (87.4)	
Multi	7205 (15.8)	1994 (7.2)	9199 (12.6)	
Multiple pregnancy				<0.001
Singleton	44,022 (96.6)	25,075 (90.6)	69,097 (94.3)	
Multiple	1572 (3.4)	2612 (9.4)	4184 (5.7)	
Adnexal surgery				<0.001
Absent	44,160 (96.9)	27,267 (98.5)	71,427 (97.5)	
Present	1434 (3.1)	420 (1.5)	1854 (2.5)	
Uterine leiomyoma				<0.001
Absent	42,296 (92.8)	26,669 (96.3)	68,965 (94.1)	
Present	3298 (7.2)	1018 (3.7)	4316 (5.9)	
Endometriosis				<0.001
Absent	43,925 (96.3)	27,174 (98.1)	71,099 (97)	
Present	1669 (3.7)	513 (1.9)	2182 (3)	
Obesity				0.25
Absent	45,532 (99.9)	27,640 (99.8)	73,172 (99.9)	
Present	62 (0.1)	47 (0.2)	109 (0.1)	

CCI—Charlson comorbidity index; SES—socioeconomic status; PCOS—polycystic ovary syndrome. Data are expressed as the number (%) or median [25 percentile, 75 percentile].

**Table 2 jcm-12-01738-t002:** Incidence of GDM or PIH in women with and without PCOS, 2012–2020, from the National Health Insurance database.

	Non-PCOS	PCOS	Total	*p*-Value	RR (95% CI)
Number of women	45,594	27,687	73,281		
GDM				<0.001	1.709 (1.61–1.814)
Absent	43,273 (94.9)	25,362 (91.6)	68,635 (93.7)		
Present	2321 (5.1)	2325 (8.4)	4646 (6.3)		
PIH				0.016	1.385 (1.062–1.808)
Absent	45,475 (99.7)	27,587 (99.6)	73,062 (99.7)		
Present	119 (0.3)	100 (0.4)	219 (0.3)		

GDM—gestational diabetes mellitus; PCOS—polycystic ovary syndrome; PIH—pregnancy-induced hypertension; RR—relative risk. Data are expressed as the number (%).

**Table 3 jcm-12-01738-t003:** Relative risks for gestational diabetes or pregnancy-induced hypertension in women with and without PCOS.

	GDM		PIH	
	RR (95% CI) ^a^	*p*-Value	RR (95% CI) ^a^	*p*-Value
Adjusted ^a^				
PCOS	1.719 (1.616–1.828)	<0.001	1.243 (0.94–1.644)	0.127
Age at inclusion (years) (reference = 20~24)				
25~29	1.585 (1.391–1.806)	<0.001	1.209 (0.686–2.13)	0.511
30~34	1.896 (1.667–2.157)	<0.001	1.56 (0.897–2.714)	0.115
35~39	2.551 (2.204–2.953)	<0.001	1.642 (0.861–3.131)	0.132
40~44	3.431 (2.646–4.45)	<0.001	4.677 (1.854–11.8)	0.001
44~49	7.681 (2.153–27.408)	0.002	0 (0-Infinite)	0.969
Low SES	1.131 (0.709–1.804)	0.607	2.727 (0.661–11.246)	0.165
Rural area	1.117 (1.051–1.188)	<0.001	0.868 (0.656–1.149)	0.322
CCI (reference = 0)				
1	1.1 (1.005–1.205)	0.119	1.866 (1.328–2.622)	0.008
≥2	1.012 (0.872–1.174)	0.643	1.022 (0.521–2.005)	0.401
Primiparity	1.447 (1.304–1.605)	<0.001	2.293 (1.292–4.082)	0.005
Multiple pregnancy	1.127 (1.004–1.265)	0.043	3.668 (2.605–5.165)	<0.001
Adnexal surgery before inclusion	0.995 (0.803–1.232)	0.96	0.719 (0.296–1.744)	0.466
Uterine leiomyoma	0.965 (0.846–1.1)	0.592	0.751 (0.417–1.353)	0.34
Endometriosis	0.841 (0.686–1.031)	0.096	2.399 (1.31–4.395)	0.005
Obesity	0.958 (0.444–2.07)	0.913	2.626 (0.354–19.478)	0.345

CCI—Charlson comorbidity index; CI—confidence interval; GDM—gestational diabetes mellitus; PCOS—polycystic ovary syndrome; PIH—pregnancy-induced hypertension; RR—relative risk; SES—socioeconomic status. ^a^ RRs were adjusted for PCOS, age, SES, region, CCI, multipara, multiple pregnancies, adnexal surgery, uterine leiomyoma, endometriosis, obesity, and (PIH or GDM).

## Data Availability

Due to the Health Insurance Review and Assessment Service (HIRA)’s privacy policy, only authorized researchers can access the data.

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
