# Peer review of "Risk of Gestational Diabetes and Pregnancy-Induced Hypertension with a History of Polycystic Ovary Syndrome: A Nationwide Population-Based Cohort Study"

_jcm, 2023, doi:10.3390/jcm12051738_

Round 1
Reviewer 1 Report
Article review (Risk of gestational diabetes and pregnancy-induced hypertension with a history of polycystic ovary syndrome: a nationwide population-based cohort study):
Summary: This is a retrospective, case-control study evaluating the association of polycystic ovary syndrome (PCOS) with pregnancy-induced hypertension ( PIH) and gestational diabetes( GDM) using data from Korea’s National Health Insurance Service. The authors found that PCOS is associated with an increased risk for GDM but not PIH. Overall, this is an interesting study that could contribute to the available literature on predisposing factors for GDM.
Comments/Revisions
1. As the authors recognize, obesity is a significant potential confounding factor, and it seems that it was not accurately documented in this dataset. The authors should provide the specific BMI cutoff used to define obesity in this database.
2. There are a few phrasing errors throughout the manuscript. Professional editing is recommended before publication.
Recommendation
Minor revision
Author Response
Please read file.

Reviewer 2 Report
An interesting study, however it would have been nice with information on BMI and body composition in the two groups.
1. Do we know anything about the quality of Koeran registries. Have they been validated in previous studies?
2. There was a higher rate of multiple pregnancies in the PCOS group - did you perform analyses where you excluded this group as twin pregnancies are associated with higher risk of GDM and PIH?
Author Response
Please read file.

Reviewer 3 Report
MATERIALS AND METHODS
Line 110: please clarify the meaning of the acronym “SES”
DISCUSSION
It would be appropriate to discuss the literature on the pathophysiological basis of PCOS and the relationship with pregnancy complications
Author Response
Please read file.
